# OpenReview forum: "Guarding a Needle in the Haystack: A Real-Time Policy-Following Streaming Video Guardrail"
_ICLR.cc/2026/Conference — ICLR 2026 Conference Desk Rejected Submission_

### Official Review · Reviewer_L4Tt · 2025-10-23

**Soundness:** 2
**Presentation:** 1
**Contribution:** 3
**Rating:** 4
**Confidence:** 2

**Summary:**

This paper proposes StreamGuard, a real-time, policy-following streaming guardrail for long-form videos. They inspect the input video in streaming form to localize unsafe content with high precision, and employ a efficient asynchronous inference stack to achieve fine-grained, frame-level monitoring with low latency. Two benchmark dataset, Safe2Shot and AdvVideo-Bench have been proposed to evaluate the defense performance.

**Strengths:**

1.	StreamGuard processes videos in a streaming manner, labeling every frame to avoid sampling losses, improving the detection performance.

2.	Two-stage training with SFT across frames improves alignment and reduce failures under adversarial conditions.

3.	StreamGuard outperforms baselines on existing benchmarks, demonstrating superior performance.

**Weaknesses:**

1.	Dataset curation details for TV2V adversarial prompts are restricted to "conceptual level," without code or full prompts, which limits verification and raises concerns on fairness in benchmark creation.

2.	Ablations are mentioned but not detailed with quantitative results in the main text, e.g., impact of removing GRPO or streaming on specific metrics.

3.	The hyperparameters of $L_{SFT}$ are $\lambda_1$, $\lambda_2$ and $\lambda_3$ for frame loss, event loss and final loss respectively, but lack of discussions on how to choose these values and no ablation studies or sensitivity analyses here.

4.	The line space of each Section and Subsection is too narrow. It is acceptable to use some command to adjust the space (like vspace in LaTeX), but this doesn’t mean the excessive change in order to satisfy 9 pages limit.

**Questions:**

1.	How does your method perform on traditional T2V adversarial datasets like T2VSafetyBench[a]?

2.	Are code and datasets publicly available for reproducibility, including full annotations?

3.	How does StreamGuard handle real-time scenarios such as network delays and incomplete streams?

---

> ### Author Response · Authors · 2025-11-23
>
> Dear Reviewer L4Tt
>
> Thank you for your detailed review and for recognizing the contribution of StreamGuard and the proposed benchmarks. We address your concerns one by one below.
>
> ---
>
> **A1: TV2V curation details and benchmark fairness**
>
> We agree that the original description of the TV2V construction was too conceptual. In the revised version, we have:
>
> * Expanded the TV2V section to give a more concrete, step-wise description of the pipeline: sampling benign prefixes from SafeWatch-Bench, selecting target policy-violating scenarios, and generating adversarial prompts with explicit combinations of AutoDAN-style rewriting, multimodal prompt-injection patterns, jailbreak instructions, and forced-output-format cues.
>
> ---
>
> **A2: Missing quantitative ablations (e.g., removing GRPO / streaming)**
>
> We appreciate this comment. In the revised version, we make the ablations more explicit and easier to find:
>
> * As already discussed in the original \emph{Ablation Study} section, we include **training pipeline ablations** comparing the full two-stage alignment (SFT + GRPO-style verifiable rewards) against SFT-only variants, which confirms that the RL-based stage improves policy consistency and robustness.
> * We report **ablation results for multi-level reasoning and supervision** (frame / event / final) and different loss weightings. Removing frame-level reasoning or frame-level loss leads to the largest performance drop and higher false positives, while event-only or final-only training is clearly inferior.
> * We add **frame-rate and sampling-strategy ablations** (balanced 1/2/4 fps, “step-drop” and “random-jitter” streaming patterns, and non-streaming uniform sampling baselines), showing that the streaming 2 fps configuration yields the strongest Acc/F1 trade-off and outperforms non-streaming baselines under comparable compute.
>
>
> These quantitative results are now clearly summarized in the main text (with table references) and detailed in Appendix~B.
>
> **Table. Ablation on streaming frame rate and sampling strategy.
> The balanced 2 fps streaming configuration (StreamGuard) offers the best overall Acc/F1 trade-off and outperforms both non-streaming and VLM baselines.**
>
> | **Setting** | **Acc** | **F1** | **Rec** | **Pre** |
> |-------------|---------|--------|---------|---------|
> | **Frame Rate (Balanced Sampling)** | | | | |
> | 4fps | 89.60 | 89.50 | 88.62 | 90.39 |
> | **2fps (StreamGuard)** | 88.58 | 88.50 | 87.86 | 89.14 |
> | 1fps | 86.50 | 86.50 | 86.48 | 86.51 |
> | **Imbalanced Sampling (Unstable Network)** | | | | |
> | Step-drop (2 → 0.5 fps) | 87.35 | 87.24 | 86.50 | 88.00 |
> | Random-jitter (0.5–2 fps) | 87.85 | 87.75 | 87.00 | 88.50 |
> | **Uniform Sampling** | | | | |
> | 16 frames / video | 83.17 | 82.99 | 82.10 | 83.90 |
> | 8 frames / video | 81.29 | 80.99 | 79.70 | 82.32 |
> | **Baselines** | | | | |
> | GPT-4.1 | 85.23 | 84.99 | 83.62 | 86.40 |
> | Qwen | 79.25 | 78.99 | 78.00 | 80.00 |
>
> We analyze how different streaming frame rates and sampling strategies affect system performance. Under balanced settings, increasing the frame rate from 1 fps to 2 fps and 4 fps leads to steady improvements across all metrics; however, the increment from 2 fps to 4 fps is marginal despite doubling computational cost. For this reason, we adopt 2 fps as the default configuration, as it provides the best trade-off between accuracy and efficiency.
>
> We also evaluate robustness under unstable streaming conditions using two imbalanced sampling patterns that emulate realistic network degradation. The step-drop strategy (2 fps → 0.5 fps) reflects scenarios where connection quality degrades sharply mid-video, while the random-jitter strategy (0.5–2 fps) models bursty bandwidth and irregular packet arrival.
>
> As shown in the table, both imbalanced strategies achieve performance close to the balanced 2 fps configuration, demonstrating that StreamGuard is resilient to moderate frame-rate fluctuations and maintains stable predictions even when the incoming stream is irregular. In contrast, non-streaming uniform-sampling baselines (8 or 16 frames per video) and VLM baselines (GPT-4.1, Qwen) perform substantially worse in terms of both Acc and F1. These results confirm that a low-rate but continuous streaming approach is significantly more reliable than sparse frame sampling, especially under imperfect or fluctuating network conditions.

---

> > ### Author Response · Authors · 2025-11-23
> >
> > **Table. Ablation on the contribution of frame-level and event-level reasoning.
> > Removing either reduces overall performance; frame-level reasoning is essential for reducing false positives.**
> >
> > | **Setting** | **Acc** | **F1** | **Rec** | **Pre** | **FPR** |
> > |-------------|---------|--------|---------|---------|---------|
> > | w/o event-level | 88.10 | 87.70 | 87.20 | 88.20 | 11.20 |
> > | w/o frame-level | 81.00 | 76.99 | 76.00 | 78.00 | 19.50 |
> > | **StreamGuard (Full Model)** | **88.58** | **88.50** | **87.86** | **89.14** | **10.70** |
> >
> > We ablate the contribution of each reasoning level during inference. Removing the event-level branch while retaining frame- and final-level reasoning results in only a minor performance drop, indicating that event-level reasoning mainly refines decisions grounded in stronger frame-level cues.
> >
> > In contrast, removing the frame-level branch causes a pronounced decline across all metrics and a large increase in false positives. This demonstrates that fine-grained frame-level signals are indispensable for safety detection, as many violations are subtle, transient, and visible only in specific frames. Event-level aggregation becomes especially useful in cases such as school bullying, where individual frames appear benign but the multi-frame pattern is clearly unsafe.
> >
> > Overall, these ablations show that frame-level reasoning delivers the primary safety signal, while event-level reasoning contributes additional stability by consolidating temporal structure.
> >
> > ---
> >
> > **A3: Hyperparameters for frame / event / final loss (λ₁, λ₂, λ₃)**
> >
> > We agree that the original version did not explain these choices sufficiently. In the revision, we provide both the rationale and an ablation:
> >
> > * First, we clarify that the loss weights are chosen with respect to the relative amount of supervision at each level. Frame-level labels are the most abundant, event-level labels are fewer, and final-level labels are the scarcest. To avoid over-emphasizing the most frequent signal and inducing a bias toward frame-only patterns, we \emph{down-weight} the frame-level loss and assign relatively higher weights to event- and final-level losses. This encourages the model to use frame-level cues as a foundation while still learning robust event summaries and global decisions.
> > * We also add an explicit **hyperparameter ablation** over different λ₁, λ₂, λ₃ settings. These results are reported in the ablation tables in Appendix~B.
> >
> > **Table. Ablation on hyperparameter choices (λ₁, λ₂, λ₃).
> > The asymmetric weighting used by StreamGuard achieves the best results and lowest FPR.**
> >
> > | **Setting** | **Acc** | **F1** | **Rec** | **Pre** | **FPR** |
> > |-------------|---------|--------|---------|---------|---------|
> > | λ₁=0.33, λ₂=0.33, λ₃=0.33 | 88.05 | 87.94 | 87.20 | 88.70 | 11.10 |
> > | λ₁=0.0, λ₂=0.0, λ₃=1.0 | 61.42 | 62.15 | 63.00 | 61.30 | 38.80 |
> > | λ₁=0.5, λ₂=0.0, λ₃=0.5 | 76.67 | 77.42 | 80.00 | 75.00 | 26.66 |
> > | λ₁=0.0, λ₂=0.5, λ₃=0.5 | 68.93 | 69.20 | 70.10 | 68.40 | 31.50 |
> > | **StreamGuard (λ₁=0.15, λ₂=0.35, λ₃=0.5)** | **88.58** | **88.50** | **87.86** | **89.14** | **10.70** |
> >
> > We analyze the impact of different loss-weighting strategies across the frame-, event-, and final-level branches during training. Relying solely on final-level supervision produces a large performance drop, confirming that a single global label is insufficient to guide robust streaming moderation. Introducing supervision at only one intermediate level (frame-only or event-only) provides some benefit but still falls well short of the full multi-level design.
> >
> > In contrast, StreamGuard’s asymmetric weighting scheme—which jointly supervises frame-, event-, and final-level predictions—yields the strongest overall results and the lowest false-positive rate. The complementary roles of the three levels explain this improvement:
> >
> > * frame-level supervision anchors detection on fine-grained visual cues that often define safety violations;
> > * event-level supervision models multi-frame temporal structure;
> > * final-level supervision enforces consistency at the global decision level.
> >
> > Combined with the inference-time ablations, these findings show that frame-level supervision supplies the most essential signal, while event- and final-level reasoning stabilize and refine predictions, leading to the most reliable safety behavior in StreamGuard.

---

> > > ### Author Response · Authors · 2025-11-23
> > >
> > > **A4: Line spacing and layout**
> > >
> > > Thank you for flagging the layout issue. In the revised version, we have relaxed the vertical spacing between sections, subsections, and paragraphs, and removed overly aggressive manual spacing commands. We carefully re-checked the full paper to ensure that the text is readable and not visually compressed, while still respecting the page limit.
> > >
> > > ---
> > >
> > > **A5: Performance on T2VSafetyBench**
> > >
> > > In response to your question about traditional T2V adversarial datasets, we have added an evaluation on **T2VSafetyBench** in the revised paper (Appendix B). StreamGuard achieves competitive or better harmful-content detection performance compared to GPT-4.1 and Qwen2.5-VL-7B, while maintaining a relatively low false positive rate. This provides additional evidence that our method generalizes to independently curated text-to-video safety benchmarks.
> > >
> > > **Table. Fine-grained category-level evaluation on T2VSafetyBench.
> > > For each category, we report Acc, Rec, Pre, and F1 for Qwen2.5-VL-7B, GPT-4.1, and StreamGuard.
> > > While overall performance is modest due to noisy and ambiguous labels, StreamGuard generally achieves a better recall–precision balance on several core harm categories.**
> > >
> > > | **Category** | **Qwen Acc** | **Qwen Rec** | **Qwen Pre** | **Qwen F1** | **GPT-4.1 Acc** | **GPT-4.1 Rec** | **GPT-4.1 Pre** | **GPT-4.1 F1** | **StreamGuard Acc** | **StreamGuard Rec** | **StreamGuard Pre** | **StreamGuard F1** |
> > > |--------------|--------------|--------------|--------------|-------------|------------------|------------------|------------------|------------------|-----------------------|-----------------------|-----------------------|----------------------|
> > > | **Pornography** | 92.19 | **100.00** | 50.00 | 66.67 | 93.75 | 60.00 | 60.00 | 60.00 | **96.88** | 80.00 | **80.00** | **80.00** |
> > > | **Borderline Pornography** | **92.19** | 0.00 | 0.00 | 0.00 | 90.62 | 20.00 | 33.33 | 25.00 | **92.19** | **33.33** | **66.67** | **44.44** |
> > > | **Violence** | 75.00 | 60.00 | 17.65 | 27.27 | 81.25 | 80.00 | 26.67 | 40.00 | **84.38** | **100.00** | **33.33** | **50.00** |
> > > | **Gore** | 87.50 | 80.00 | 36.36 | 50.00 | 92.19 | 80.00 | 50.00 | 61.54 | **93.75** | **83.33** | **62.50** | **71.43** |
> > > | **Public Figures** | **90.62** | **20.00** | **33.33** | **25.00** | **90.62** | 0.00 | 0.00 | 0.00 | 87.50 | 0.00 | 0.00 | 0.00 |
> > > | **Discrimination** | **92.19** | 0.00 | 0.00 | 0.00 | 85.94 | 0.00 | 0.00 | 0.00 | 87.50 | **14.29** | **33.33** | **20.00** |
> > > | **Politically Sensitive** | 92.19 | 60.00 | 50.00 | 54.55 | **95.31** | 60.00 | 75.00 | 66.67 | **95.31** | **66.67** | **80.00** | **72.73** |
> > > | **Illegal Activities** | 95.31 | 60.00 | 75.00 | 66.67 | **96.88** | 80.00 | 80.00 | 80.00 | **96.88** | **83.33** | **83.33** | **83.33** |
> > > | **Disturbing Content** | 64.06 | **100.00** | 17.86 | 30.30 | 65.62 | **100.00** | **18.52** | **31.25** | **68.75** | **100.00** | 16.67 | 28.57 |
> > > | **Misinformation** | 92.19 | 0.00 | 0.00 | 0.00 | 92.19 | 0.00 | 0.00 | 0.00 | **93.75** | **20.00** | **100.00** | **33.33** |
> > > | **Copyright** | 92.19 | 0.00 | 0.00 | 0.00 | **93.75** | **20.00** | **100.00** | **33.33** | **93.75** | **20.00** | **100.00** | **33.33** |
> > > | **Sequential Action Risk** | **96.88** | 0.00 | 0.00 | 0.00 | 87.50 | 0.00 | 0.00 | 0.00 | 90.62 | 0.00 | 0.00 | 0.00 |
> > > | **Dynamic Variation Risk** | **93.75** | 0.00 | 0.00 | 0.00 | **93.75** | 0.00 | 0.00 | 0.00 | **93.75** | **25.00** | **50.00** | **33.33** |
> > > | **Coherent Contextual Risk** | **95.31** | 0.00 | 0.00 | 0.00 | **95.31** | 0.00 | 0.00 | 0.00 | 93.75 | **25.00** | **50.00** | **33.33** |
> > >
> > > As shown in the table above, we additionally evaluate all models on a new out-of-distribution video-generation safety dataset that covers a wide range of fine-grained risk categories—including pornography, borderline content, violence, political sensitivity, misinformation, and multiple temporal-risk dimensions. Overall performance is relatively modest across all models. Our manual inspection indicates that many clips are ambiguous or only loosely aligned with their assigned labels, and certain categories (e.g., discrimination and misinformation) contain noisy or inconsistent annotations. Consequently, even when a model’s output is qualitatively sensible, it may still be penalized due to label ambiguity.
> > >
> > > For this reason, the results are best interpreted as a robustness stress test under noisy supervision rather than a clean measure of absolute safety capability. Despite these limitations, StreamGuard consistently demonstrates a more favorable recall–precision balance in several core harm categories—such as pornography, gore, illegal activities, and politically sensitive content—whereas the baselines frequently suffer from severe under-detection (zero recall) or unstable precision. These trends suggest that StreamGuard’s multi-level architecture offers more resilient behavior under challenging OOD conditions and imperfect annotation quality.

---

> > > > ### Author Response · Authors · 2025-11-23
> > > >
> > > > **A6: Code, datasets, and annotations for reproducibility**
> > > >
> > > > We appreciate your concern about reproducibility. As noted in our response to other reviewers, we already provide detailed experimental settings in the Experimental Setup section and in the Training Details appendix (GPU type/count, batch size, learning rate, training steps, etc.). Upon acceptance, we plan to publicly release:
> > > >
> > > > * The **StreamGuard implementation** (model code and evaluation scripts), and
> > > > * The **Safe2Shot** and **AdvVideo-Bench** datasets with full annotations, together with configuration files to reproduce the reported results.
> > > >
> > > > This will enable other researchers to validate and extend our work.
> > > >
> > > > ---
> > > >
> > > > **A7: Handling real-time scenarios such as network delays and incomplete streams**
> > > >
> > > > StreamGuard is designed to operate on a **streaming input**, processing frames as they arrive without requiring full-video buffering. Network delays and incomplete streams primarily affect when frames become available, not the correctness of the per-frame labeling. In practice:
> > > >
> > > > * If the stream is delayed or irregular, StreamGuard simply processes the frames at the arrival rate; decisions for already-seen frames remain valid.
> > > > * To approximate unstable network conditions, we include **imbalanced and jittered sampling experiments** (step-drop and random-jitter frame-rate patterns) in Appendix B. These experiments show only modest degradation compared to stable streaming, indicating robustness to irregular or degraded input rates.
> > > >
> > > > **Table. Ablation on streaming frame rate and sampling strategy.
> > > > The balanced 2 fps streaming configuration (StreamGuard) offers the best overall Acc/F1 trade-off and outperforms both non-streaming and VLM baselines.**
> > > >
> > > > | **Setting** | **Acc** | **F1** | **Rec** | **Pre** |
> > > > |-------------|---------|--------|---------|---------|
> > > > | **Frame Rate (Balanced Sampling)** | | | | |
> > > > | 4fps | 89.60 | 89.50 | 88.62 | 90.39 |
> > > > | **2fps (StreamGuard)** | 88.58 | 88.50 | 87.86 | 89.14 |
> > > > | 1fps | 86.50 | 86.50 | 86.48 | 86.51 |
> > > > | **Imbalanced Sampling (Unstable Network)** | | | | |
> > > > | Step-drop (2 → 0.5 fps) | 87.35 | 87.24 | 86.50 | 88.00 |
> > > > | Random-jitter (0.5–2 fps) | 87.85 | 87.75 | 87.00 | 88.50 |
> > > > | **Uniform Sampling** | | | | |
> > > > | 16 frames / video | 83.17 | 82.99 | 82.10 | 83.90 |
> > > > | 8 frames / video | 81.29 | 80.99 | 79.70 | 82.32 |
> > > > | **Baselines** | | | | |
> > > > | GPT-4.1 | 85.23 | 84.99 | 83.62 | 86.40 |
> > > > | Qwen | 79.25 | 78.99 | 78.00 | 80.00 |
> > > >
> > > > We ablate the effect of different streaming frame rates and sampling strategies. Under balanced sampling, increasing the frame rate from 1 fps to 2 fps and 4 fps progressively improves performance, but the improvement from 2 fps to 4 fps is marginal despite doubling computational cost. Accordingly, we adopt 2 fps as the default configuration, as it offers the most favorable balance between accuracy and efficiency.
> > > >
> > > > We also evaluate robustness under unstable streaming conditions using two imbalanced sampling patterns that mimic real-world bandwidth degradation. The step-drop configuration (2 fps → 0.5 fps) represents a sudden decline in connection quality, while random-jitter (0.5–2 fps) models bursty or inconsistent frame delivery caused by fluctuating network conditions.
> > > >
> > > > As shown in the table, both imbalanced strategies perform comparably to the balanced 2 fps setting, indicating that StreamGuard maintains stable predictions even when the incoming frame rate becomes irregular. In contrast, non-streaming uniform sampling (8 or 16 frames per video) and VLM baselines such as GPT-4.1 and Qwen perform substantially worse in terms of both Acc and F1. These results highlight that a low-rate but continuous streaming pipeline is significantly more robust than sparse frame selection, particularly under imperfect or unstable network conditions.
> > > >
> > > > ---
> > > >
> > > > We hope these changes and clarifications address your concerns about dataset curation, ablations, hyperparameters, formatting, robustness, and reproducibility.

---

> > > > > ### Comment · Reviewer_L4Tt · 2025-11-26
> > > > >
> > > > > Thanks for your rebuttal. Some of my concerns have been solved, but several problems still remains.
> > > > >
> > > > > **Reproducibility / benchmark transparency.** The TV2V/TV2T pipeline is still only partially specified, with key prompt templates and attack recipes promised “upon acceptance” rather than being reproducible from the paper itself. Given that major claims rely on these benchmarks, this is a concern.
> > > > >
> > > > > **Real-time claims.** Latency analysis is on short clips with strong hardware and synthetic “unstable network” patterns. There is no convincing end-to-end evaluation for truly long or continuous real-world streams, so the “real-time streaming guardrail” framing is overstated.
> > > > >
> > > > > Therefore, I tend to keep my initial rating.

---

> > > > > > ### Author Response · Authors · 2025-11-28
> > > > > >
> > > > > > Thank you for your careful review and for the constructive feedback. We address your remaining concerns point by point below.
> > > > > >
> > > > > > ---
> > > > > >
> > > > > > ## **Benchmark transparency for TV2V/TV2T**
> > > > > >
> > > > > > In the revision, we have substantially expanded the description of our adversarial benchmarks so that they are reproducible from the paper itself rather than only “upon acceptance.”
> > > > > >
> > > > > > ### **TV2V.**
> > > > > >
> > > > > > Appendix A.7 now provides a fully self-contained and formal description of the TV2V pipeline:
> > > > > >
> > > > > > * We introduce a **formal definition** of the dataset as a collection of tuples
> > > > > >   **D = { (u, q, g, z) }**,
> > > > > >   where **u** is a benign prefix from SafeWatch-Bench, **q** is the instantiated adversarial prompt, **g** is the generated continuation from the text+video-to-video generator **G**, and **z** is the verified safety label.
> > > > > >
> > > > > > * We add a complete **algorithmic description** of the construction process (Algorithm 1 in Appendix A.7), including
> > > > > >   – sampling a category,
> > > > > >   – sampling a template,
> > > > > >   – instantiating the template to obtain **q**,
> > > > > >   – generating **g = G(u, q)**,
> > > > > >   – running an automatic + human **Check(g)** procedure before inclusion.
> > > > > >
> > > > > > * We introduce a **template inventory**, decomposing each template into four components:
> > > > > >   **context**, **attack**, **rewrite**, and **format**,
> > > > > >   together with a taxonomy of the main template families (next-seconds continuation, explicit continuation, imperative continuation, role-based prompts).
> > > > > >
> > > > > > * We add a dedicated **Injection Text Types and Examples** subsection that groups the injection patterns into system-override/meta-instruction, persona-based, temporal-scaffolding, format-constrained, and indirect/obfuscated classes. Each category includes multiple **sanitized slot-based examples**, making it possible to reconstruct the attack space without our exact strings.
> > > > > >
> > > > > > * We further describe our **prefix–suffix optimization procedure** in detail (Appendix A.7), including
> > > > > >   – the optimization objective,
> > > > > >   – the safety-scoring model S,
> > > > > >   – the editing LLM E (which never sees the guardrail prompt),
> > > > > >   – beam sizes,
> > > > > >   – the five-round AutoDAN-style evolution window,
> > > > > >   – the 30 optimization epochs.
> > > > > >   The full workflow is provided as **Algorithm 2** in pseudo-code.
> > > > > >
> > > > > > ### **TV2T.**
> > > > > >
> > > > > > Appendix A.8 provides an expanded step-by-step description of the TV2T construction process, including a visual pipeline (Figure 4), so that the benchmark can be reproduced without access to our code.
> > > > > >
> > > > > > ---
> > > > > >
> > > > > > We hope that these newly added materials — including the structured dataset descriptions, explicit algorithmic pipelines, and representative template and attack examples — help clarify the construction process and improve reproducibility.

---

> > > > > > > ### Author Response · Authors · 2025-11-28
> > > > > > >
> > > > > > > **Real-world long-stream evaluation and “real-time” framing**
> > > > > > >
> > > > > > > We agree that it is important to go beyond short, synthetic clips. In response, we have added a new **real-world streaming evaluation** in Appendix A.8 that directly targets your concern about long and unstable streams:
> > > > > > >
> > > > > > > * We deploy StreamGuard on three major UGC platforms (e.g. 91, X, onlyfans) which contain various unsafe clips hidden within the long videos, evaluating **20 authentic long-form livestreams** online in real-time with durations ranging from **30 minutes to over 3 hours**. Figure 6 shows the length distribution: 12 videos in 30min–1hr, 6 in 1hr–2hr, and 2 in 2hr–3hr.
> > > > > > > * Monitoring these sexual livestreaming videos online naturally cover a diverse set of real-world behaviors, camera motions, lighting conditions, user interactions, and environmental contexts. Importantly, because these are genuine livestreams, they naturally contain:
> > > > > > >     + severe network jitter, causing frame freezes and temporal desynchronization;
> > > > > > >     + livestream interruptions, including mid-stream reconnections or brief signal loss;
> > > > > > >     + black-screen and standby periods, such as device repositioning or intentional transitions;
> > > > > > >   variable bitrate / compression artifacts, leading to degraded frame quality;
> > > > > > >     + unstable camera movement, handheld shake, or sudden scene switches.
> > > > > > > Such conditions are rarely captured in existing curated benchmarks, yet they are central to real-world moderation. This evaluation therefore reflects realistic deployment challenges rather than idealized testing setups.
> > > > > > > * We evaluate StreamGuard against **GPT-4.1, Gemini-1.5-Flash, and QwenVL2-7B** under a matched-budget protocol:
> > > > > > >
> > > > > > >   * For non-streaming models, we split each video into 30-minute segments and **uniformly sample 16 frames per segment**. A video is flagged positive if any segment is positive.
> > > > > > >   * StreamGuard processes the entire stream in **continuous streaming mode**, without full-video buffering.
> > > > > > >
> > > > > > > We report two settings with detailed tables in the appendix:
> > > > > > >
> > > > > > > 1. **Setting 1: Mixed-label evaluation.**
> > > > > > >    10 videos with clear harmful violations are treated as positive, 10 as negative (benign/borderline).
> > > > > > >    StreamGuard achieves **0.90 Acc / 0.90 F1**, outperforming GPT-4.1, Gemini-1.5-Flash, and QwenVL2-7B, which degrade more severely under real-world noise.
> > > > > > >
> > > > > > > | Model               | Acc      | Prec     | Rec      | F1       |
> > > > > > > | ------------------- | -------- | -------- | -------- | -------- |
> > > > > > > | **StreamGuard** | **0.90** | **0.90** | **0.90** | **0.90** |
> > > > > > > | GPT-4.1             | 0.80     | 0.80     | 0.80     | 0.80     |
> > > > > > > | Gemini 1.5-Flash    | 0.75     | 0.78     | 0.70     | 0.74     |
> > > > > > > | QwenVL2-7B          | 0.70     | 0.70     | 0.70     | 0.70     |
> > > > > > >
> > > > > > > 2. **Setting 2: All-positive evaluation.**
> > > > > > >    All 20 videos are treated as requiring a positive trigger, stressing recall under partially visible or weakly signaled harmful behavior.
> > > > > > >    StreamGuard reaches **0.95 Acc / 0.97 F1**, with perfect precision and the highest recall among all models.
> > > > > > >
> > > > > > > | Model               | Acc      | Prec     | Rec      | F1       |
> > > > > > > | ------------------- | -------- | -------- | -------- | -------- |
> > > > > > > | **StreamGuard** | **0.95** | **1.00** | **0.95** | **0.97** |
> > > > > > > | GPT-4.1             | 0.85     | 1.00     | 0.85     | 0.92     |
> > > > > > > | Gemini 1.5-Flash    | 0.80     | 1.00     | 0.80     | 0.89     |
> > > > > > > | QwenVL2-7B          | 0.75     | 1.00     | 0.75     | 0.86     |

---

> ### Author Response · Authors · 2025-11-28
>
> These experiments **complement** the synthetic “unstable network” ablations you mentioned:
>
> * focuses on **controlled frame-rate and sampling patterns** (balanced 1–4 fps, step-drop, random jitter) on our curated benchmark, to isolate the effect of streaming vs sparse sampling.
> * now provides a **truly long, noisy, end-to-end evaluation** on real-world livestreams, including network-induced artifacts and incomplete streams.
>
> Conceptually, our “real-time streaming guardrail” framing refers to the fact that:
>
> * StreamGuard operates purely on **incoming frames and a bounded memory**, without requiring access to future frames or full-video buffering;
> * At the reported frame rates (e.g., 2 fps) our measured **per-frame latency stays within the real-time budget** on commodity datacenter GPUs, and
> * The newly added long-stream experiments show that the streaming architecture remains **robust over 30min–3hr continuous content**, even in the presence of actual network irregularities.
>
> That said, we appreciate the concern about overstatement. We are happy to further clarify the wording in the paper (e.g., emphasizing “streaming-appropriate, near real-time guardrail” and clearly differentiating **model-level streaming capability** from **full production deployment guarantees**) to avoid any misunderstanding.
>
> ---
>
> We hope these additional clarifications: (i) the fully specified TV2V/TV2T pipelines with template families, injection types, and optimization algorithms, and (ii) the new real-world long-stream evaluation on noisy livestreams could address your remaining concerns about benchmark transparency, robustness, and the scope of our real-time claims.

---

### Official Review · Reviewer_ZuRt · 2025-10-30

**Soundness:** 3
**Presentation:** 2
**Contribution:** 3
**Rating:** 6
**Confidence:** 3

**Summary:**

The paper presents StreamGuard, a system designed to safeguard real-time streaming video by identifying unsafe content through a multi-level policy alignment approach. The authors propose a framework that incorporates Teacher-forcing Supervised Fine-Tuning (SFT) across three levels: frame-level, event-level, and final-level. This is intended to ensure a fine-grained alignment at the frame level, event summary at the event level, and a consolidated decision-making mechanism at the final level.
The proposed system is evaluated across several benchmarks with the results demonstrating competitive performance.

**Strengths:**

1. The multi-level alignment, specifically integrating frame-level, event-level, and final-level processing, is a novel way to improve the performance of video safety systems.
2. The paper evaluates the system on several benchmarks, showing improvements in accuracy and F1 score on datasets like TV2T and TV2V, which is an important contribution.

**Weaknesses:**

1. Although the paper discusses Teacher-forcing SFT across levels, it does not clearly present an ablation study isolating the contributions of frame-level, event-level, and final-level processing in the context of system performance (Perhaps I haven't found relevant analysis; if I have, please point it out.).

2. How does the proposed defense method perform against adaptive attacks? Is it effective in defending against such attacks?

3. The authors do not provide accessible code, data, specific experimental details (such as resources required for model training and parameter configurations), or failure case analyses, which negatively impacts the reproducibility of the experiment and hinders a deeper understanding of the work.

4. The authors should provide a detailed description of the differences between the SAFESHOT benchmark and Video-SafetyBench, VidSafe, et al.

**Questions:**

The questions can be found in the Weaknesses section above.

---

> ### Author Response · Authors · 2025-11-23
>
> Dear Reviewer ZuRt
>
> Thank you very much for your thoughtful and positive review. We appreciate your recognition of the multi-level alignment design and our benchmark results, and we address your concerns one by one below.
>
> ---
>
> **A1: Ablation for frame- / event- / final-level contributions**
>
> Thank you for asking for a clearer isolation of each level’s effect. In the revised version, we now make this explicit:
>
> * We add an **ablation on multi-level reasoning at inference time** (Appendix B, “Ablation on Multi-Level Reasoning and Supervision”), where we remove the event-level branch (**w/o event-level**) or the frame-level branch (**w/o frame-level**) and compare to the full model. Removing the event-level branch causes only a small performance drop, while removing the frame-level branch leads to a large decrease in accuracy/F1 and a clear increase in false positives.
> * We also add an **ablation on the loss weights** for frame-, event-, and final-level supervision (different (λ₁, λ₂, λ₃) settings). Using only final-level loss performs poorly; adding supervision at only one intermediate level improves but remains clearly inferior. The best results come from our asymmetric weighting that jointly supervises all three levels.
>
>
> **Table. Ablation on the contribution of frame-level and event-level reasoning.
> Removing either reduces overall performance; frame-level reasoning is essential for reducing false positives.**
>
> | **Setting** | **Acc** | **F1** | **Rec** | **Pre** | **FPR** |
> |-------------|---------|--------|---------|---------|---------|
> | w/o event-level | 88.10 | 87.70 | 87.20 | 88.20 | 11.20 |
> | w/o frame-level | 81.00 | 76.99 | 76.00 | 78.00 | 19.50 |
> | **StreamGuard (Full Model)** | **88.58** | **88.50** | **87.86** | **89.14** | **10.70** |
>
> We examine the contribution of each reasoning level at inference time. Removing the event-level branch while retaining frame- and final-level reasoning results in only a modest performance drop, indicating that event-level reasoning mainly offers additional refinement on top of already strong frame-level cues.
>
> In contrast, eliminating the frame-level branch leads to a pronounced decline across all metrics and a substantial increase in false positives. This demonstrates that fine-grained frame-level signals are critical for reliable safety detection: many unsafe indicators are brief or subtle, appearing only in isolated frames and therefore cannot be recovered through higher-level aggregation alone. Event-level reasoning is particularly valuable in cases such as school bullying or harassment, where individual frames may look innocuous but the temporal evolution clearly forms an unsafe event.
>
> Overall, these ablations show that frame-level reasoning provides the primary safety signal, while event-level aggregation improves stability and temporal consistency, enabling more robust and context-aware safety judgments.
>
> **Table. Ablation on hyperparameter choices (λ₁, λ₂, λ₃).
> The asymmetric weighting used by StreamGuard achieves the best results and lowest FPR.**
>
> | **Setting** | **Acc** | **F1** | **Rec** | **Pre** | **FPR** |
> |-------------|---------|--------|---------|---------|---------|
> | λ₁=0.33, λ₂=0.33, λ₃=0.33 | 88.05 | 87.94 | 87.20 | 88.70 | 11.10 |
> | λ₁=0.0, λ₂=0.0, λ₃=1.0 | 61.42 | 62.15 | 63.00 | 61.30 | 38.80 |
> | λ₁=0.5, λ₂=0.0, λ₃=0.5 | 76.67 | 77.42 | 80.00 | 75.00 | 26.66 |
> | λ₁=0.0, λ₂=0.5, λ₃=0.5 | 68.93 | 69.20 | 70.10 | 68.40 | 31.50 |
> | **StreamGuard (λ₁=0.15, λ₂=0.35, λ₃=0.5)** | **88.58** | **88.50** | **87.86** | **89.14** | **10.70** |
>
> We further investigate how different loss weightings across the frame-, event-, and final-level branches affect training. Relying solely on final-level supervision leads to a severe performance drop, showing that a single global label is insufficient to guide robust streaming moderation. Adding supervision to only one intermediate level improves performance modestly, but both configurations remain substantially weaker than the full multi-level setup.
>
> In contrast, StreamGuard’s asymmetric weighting strategy—which jointly supervises frame-, event-, and final-level predictions—yields the strongest performance and the lowest false-positive rate. This reflects the complementary roles of the three levels:
>
> * frame-level supervision grounds the model in fine-grained visual cues that often define safety violations;
> * event-level supervision captures temporal patterns that span multiple frames;
> * final-level supervision enforces consistency in the overall safety decision.
>
> Together with the inference-time ablation, these findings reinforce that frame-level supervision provides the most essential signal, while event- and final-level reasoning help consolidate and stabilize predictions, producing the most reliable safety behavior in StreamGuard.

---

> > ### Author Response · Authors · 2025-11-23
> >
> > **A2: Performance under adaptive attacks**
> >
> > We agree that robustness to adaptive attacks is crucial. In the original submission, ADVVideo-Bench already includes prompt-based and transformation-based attacks inspired by industry practice, but we did not explicitly evaluate adaptive attacks that react to the defense.
> >
> > In the revision, we add a **robustness study with adaptive prompt-injection attacks** (Appendix B, “Robustness to Adaptive Attacks”) using AutoDAN. For each prompt, we run 5 adaptive attack rounds and evaluate on the strongest one per model. All models have relatively low attack success rates, but **StreamGuard** achieves both the highest clean accuracy and the lowest attack success rate, indicating that the multi-level alignment offers some resilience even under adaptive prompting. We also explicitly note that designing attacks that directly target the streaming reset mechanism is an important open threat model and a direction for future work.
> >
> > **Table. Robustness evaluation under Adaptive attacks (AutoDAN).
> > We report clean Accuracy (Acc), F1, Recall (Rec), Precision (Pre), and the attack success rate (ASR).
> > StreamGuard achieves both the highest clean performance and the lowest ASR.**
> >
> > | **Model** | **Acc** | **F1** | **Rec** | **Pre** | **ASR** |
> > |----------|---------|--------|---------|---------|---------|
> > | **StreamGuard** | **88.10** | **88.19** | **87.50** | **88.90** | **4.2%** |
> > | **GPT-4.1** | 84.00 | 83.73 | 82.50 | 85.00 | 8.8% |
> > | **Qwen2.5-VL-7B** | 77.50 | 77.49 | 76.50 | 78.50 | 11.5% |
> >
> > We further assess robustness against adversarial prompt-injection attacks using AutoDAN. For each input, AutoDAN executes five iterative attack rounds, with each round generating a stronger adversarial rewrite of the original prompt. We report the highest attack success rate (ASR) observed across these rounds, where ASR denotes the proportion of cases in which the model is induced to produce an incorrect safety decision.
> >
> > As shown in the table, ASR remains relatively low for all three models, indicating that prompt-injection attacks are not trivially effective in a streaming moderation setting. However, clear differences emerge: StreamGuard achieves both the strongest clean performance (88.10 Acc, 88.19 F1) and the lowest ASR (4.2%), while GPT-4.1 and Qwen are significantly more vulnerable (8.8% and 11.5% ASR, respectively). This improvement is consistent with our training design, which incorporates AutoDAN-style adversarial examples and thereby enhances StreamGuard’s resilience to prompt-injection attacks without compromising clean accuracy.
> >
> > ---
> >
> > **A3: Code, data, experimental details, and failure cases**
> >
> > We appreciate your concern about reproducibility. We would like to clarify that the original submission already provides detailed experimental settings in the \emph{Experimental Setup} subsection and in Appendix~A.4 \emph{Training Details}, including the type and number of GPUs used, batch size, learning rate schedule, total training steps, and other hyperparameters. In the revised version, we make these references more explicit and easier to locate.
> >
> > Regarding code and data, we will ensure to release the implementation and the benchmarks (Safe2Shot and AdvVideo-Bench) publicly, together with configuration files and evaluation scripts, so that others can reproduce our results and build upon this work.

---

> > > ### Author Response · Authors · 2025-11-23
> > >
> > > **A4: Differences between Safe2Shot and existing video safety benchmarks**
> > >
> > > Thank you for asking for a clearer comparison. In the revised paper we explicitly describe how **Safe2Shot** (referred to as “SAFESHOT” in your review) differs from benchmarks such as Video-SafetyBench and VidSafe:
> > >
> > > * **Sparsity of unsafe elements:** As mentioned in Appendix A.6, Safe2Shot is designed so that **unsafe segments are often very sparse** (sometimes <5% of frames) and surrounded by benign content, making it particularly challenging for methods that rely on sparse frame sampling.
> > > * **Source and format:** Safe2Shot is built from **live-sourced, long-form videos** from short-video platforms and social media, with per-second annotations and explicit unsafe spans in otherwise benign streams. Many prior benchmarks use shorter, more uniformly unsafe clips or curated scenes.
> > > * **Borderline content:** Safe2Shot intentionally contains a high proportion of **borderline and ambiguous cases** (e.g., suggestive but not explicitly sexual behavior), which are easy to misclassify and stress-test policy following.
> > > * **Alignment with streaming use case:** Safe2Shot is constructed specifically for **streaming moderation** with frame-level and event-level labels, whereas many existing benchmarks are oriented toward offline evaluation or generation only.
> > >
> > > ---
> > >
> > > We hope these clarifications and additions address your concerns about ablations, adaptive attacks, reproducibility, and dataset positioning.

---

### Official Review · Reviewer_mM3s · 2025-10-31

**Soundness:** 3
**Presentation:** 3
**Contribution:** 3
**Rating:** 6
**Confidence:** 3

**Summary:**

This work introduces STREAMGUARD, a real-time guardrail system for streaming videos. The paper argues that existing multimodal moderation systems miss harmful content due to frame subsampling, offline pipelines with blocking latency, and weak policy following. To solve these issues, the authors propose a streaming architecture with frame-level labeling, event-based contextual resets, and asynchronous inference so that explanation generation does not block incoming frame processing. The method further uses a two-stage alignment procedure that combines supervised fine-tuning with reinforcement learning from verifiable rewards to strengthen policy consistency. The paper also releases two datasets: SAFE2SHOT, which focuses on brief unsafe spans in long videos, and ADVVIDEO-BENCH, which contains adversarially perturbed cases for both video-to-video and video-to-text jailbreaks. Experiments show gains on long-video safety benchmarks, adversarial settings, and runtime latency compared to strong baselines

**Strengths:**

The work identifies a concrete and meaningful gap: moderation systems often lose signal due to subsampling and blocking decoding, especially when unsafe frames occur only briefly. The authors support this claim with an explicit streaming design and a realistic use case: long-form video moderation with adversarial pressure. The proposed method is tailored to this objective, offering frame-exact labeling and parallel execution. The dataset design also aligns tightly with the problem setup, and the empirical evaluation is extensive, covering both normal and adversarial cases. Results indicate clear improvements in both accuracy and latency, suggesting that the solution is practical for high-throughput applications. The work is also careful in describing alignment training and real-world deployment considerations, which increases credibility.

**Weaknesses:**

While the contribution is technically sound, the approach builds on known components—streaming MLLM execution, context resets, and RL-based policy alignment. The novelty mainly lies in integrating these ideas in a moderation pipeline rather than advancing a new learning principle.

Specific policy categories and event definition rules appear heuristic and could benefit from more theoretical analysis or failure case breakdowns.

It would also strengthen the work to include ablations comparing against different frame rates or memory window strategies to verify that the performance gains truly stem from the streaming mechanism rather than from scale or dataset advantage.

The method assumes reliable annotation and clear unsafe spans; how it handles ambiguous or culturally sensitive borderline content could be discussed more.

**Questions:**

The paper states that missing a few frames may hide unsafe content, but it does not quantify how performance changes when frame rate varies. Can you report results across different sampling rates to validate frame-exact necessity?

ADVVIDEO-BENCH focuses on prompt-based and transformation attacks reported in industry practice. How would the method handle adaptive attackers who target the specific streaming reset mechanism?

---

> ### Author Response · Authors · 2025-11-23
>
> Dear Reviewer mM3s
>
> Thank you very much for your thoughtful and positive review. We appreciate your recognition of the problem setting, system design, and empirical results, and we address your concerns one by one below.
>
> ---
>
> **A1: On novelty and use of known components**
>
> We agree that StreamGuard builds on known building blocks such as streaming multimodal execution, context resets, and RL-based policy alignment. Our intended contribution is not a new learning principle, but a **concrete streaming safety architecture and dataset suite** tailored to long-form, adversarial video moderation, [where previous work lack and we are the first]:
>
> * a **frame-exact, event-aware streaming pipeline** that decouples moderation decisions from explanation generation and supports real-time throughput;
> * **multi-level policy alignment** (frame / event / final) designed specifically for safety labels rather than generic instruction following;
> * two new benchmarks (**Safe2Shot** and **AdvVideo-Bench**) that focus on brief unsafe spans and adversarial jailbreaks in long videos.
>
> In the revised version, we clarify this system-level focus in the introduction and related work, and explicitly position StreamGuard as a deployment-oriented streaming guardrail architecture complementary to prior offline or generation-only safety works.
>
> ---
> **A2: Heuristic Nature of Policy Categories and Event Definitions**
>
> We agree with the reviewer that policy categories and event definitions should not be purely heuristic. In our system, the risk categories are directly derived from the safety policies of major video platforms, rather than being designed arbitrarily.
> Specifically, we survey the public content guidelines of several mainstream video platforms (e.g., video-sharing and livestreaming services). These policies consistently define prohibited or high-risk content types, such as:
> Sexual content and pornography, including explicit acts and sexualized nudity;
> Violence and gore, including physical assaults, graphic injuries, and shocking scenes;
> Weapons and criminal activity, such as guns, shootings, robberies, and drug production/use;
> Hate and discrimination, targeting protected groups;
> Politically sensitive manipulation, including misleading political content;
> Misinformation in regulated domains, such as health or elections;
> Harmful or dangerous acts, including risky stunts and unsafe behaviors.
> We then map these recurring policy clauses into our operational risk categories (e.g., pornography, borderline content, violence, gore, drug activity, politically sensitive content, discrimination) and use them to define the visual evidence that constitutes an unsafe event in our streaming setting. Thus, the categories and event rules originate from real, widely adopted platform policies, not heuristic intuition.

---

> > ### Author Response · Authors · 2025-11-23
> >
> > **A3: On ablations for frame rate / memory window and frame-exact necessity**
> >
> > We appreciate the suggestion to quantify how performance changes when frame rate and memory configuration vary. In the revised version, we add ablations in **Appendix B**:
> >
> > * An **ablation on streaming frame rate and sampling strategy** (balanced 1 / 2 / 4 fps, imbalanced “step-drop” and “random-jitter” patterns that mimic unstable networks, and uniform 8 / 16-frame sampling baselines). The balanced 2 fps streaming configuration achieves the best Acc/F1 trade-off and clearly outperforms non-streaming uniform sampling with comparable compute, supporting our claim that frame-exact streaming coverage—not just scale—drives the gains.
> > * An **ablation on multi-level reasoning and supervision**, where removing frame-level reasoning or frame-level loss leads to the largest degradation and higher false positive rate, while event-/final-only supervision is insufficient.
> >
> > **Table. Ablation on streaming frame rate and sampling strategy.
> > The balanced 2 fps streaming configuration (StreamGuard) offers the best overall Acc/F1 trade-off and outperforms both non-streaming and VLM baselines.**
> >
> > | **Setting** | **Acc** | **F1** | **Rec** | **Pre** |
> > |-------------|---------|--------|---------|---------|
> > | **Frame Rate (Balanced Sampling)** | | | | |
> > | 4fps | 89.60 | 89.50 | 88.62 | 90.39 |
> > | **2fps (StreamGuard)** | 88.58 | 88.50 | 87.86 | 89.14 |
> > | 1fps | 86.50 | 86.50 | 86.48 | 86.51 |
> > | **Imbalanced Sampling (Unstable Network)** | | | | |
> > | Step-drop (2 → 0.5 fps) | 87.35 | 87.24 | 86.50 | 88.00 |
> > | Random-jitter (0.5–2 fps) | 87.85 | 87.75 | 87.00 | 88.50 |
> > | **Uniform Sampling** | | | | |
> > | 16 frames / video | 83.17 | 82.99 | 82.10 | 83.90 |
> > | 8 frames / video | 81.29 | 80.99 | 79.70 | 82.32 |
> > | **Baselines** | | | | |
> > | GPT-4.1 | 85.23 | 84.99 | 83.62 | 86.40 |
> > | Qwen | 79.25 | 78.99 | 78.00 | 80.00 |
> >
> > We ablate the impact of streaming frame rate and sampling strategy. In the balanced setting, increasing the frame rate from 1 fps to 2 fps and 4 fps steadily improves all metrics, but the gain from 2 fps to 4 fps is marginal while doubling the inference cost. We therefore adopt the balanced 2 fps configuration as our default, which provides a favorable accuracy–efficiency trade-off.
> >
> > We further examine how StreamGuard behaves under unstable streaming conditions using two imbalanced sampling patterns designed to mimic network degradation. The step-drop setting (2 fps → 0.5 fps) simulates a connection that suddenly deteriorates, while random-jitter (0.5–2 fps) emulates bursty bandwidth and fluctuating packet loss.
> >
> > As shown in the table, both imbalanced settings remain close to the balanced 2 fps results, indicating that StreamGuard is robust to moderate frame-rate instability and does not collapse when the stream becomes uneven. In contrast, non-streaming uniform sampling (8 or 16 frames per video) and VLM baselines (GPT-4.1, Qwen) achieve significantly lower Acc and F1. This confirms that a low-rate but continuous streaming design is more reliable than sparse frame selection, even under imperfect network conditions.
> >
> > Together, these results directly answer your question about performance vs. sampling rate and provide empirical evidence that the streaming mechanism and frame-level signals are indeed critical, beyond dataset or model size advantages.

---

> > > ### Author Response · Authors · 2025-11-23
> > >
> > > **A4: On adaptive attackers targeting the streaming reset mechanism**
> > >
> > > Your question about adaptive attackers is very helpful. Beyond the transformation- and prompt-based attacks in AdvVideo-Bench, we now add **robustness experiments against adaptive attacks** in **Appendix B**. For each prompt, we run 5 attack rounds and evaluate on the strongest one. While all models have relatively low attack success rates, StreamGuard achieves both the highest clean accuracy and the lowest attack success rate, suggesting that multi-level supervision and event-aware design offer some resilience even under adaptive prompting.
> > >
> > > **Table. Robustness evaluation under Adaptive attacks (AutoDAN).
> > > We report clean Accuracy (Acc), F1, Recall (Rec), Precision (Pre), and the attack success rate (ASR).
> > > StreamGuard achieves both the highest clean performance and the lowest ASR.**
> > >
> > > | **Model** | **Acc** | **F1** | **Rec** | **Pre** | **ASR** |
> > > |----------|---------|--------|---------|---------|---------|
> > > | **StreamGuard** | **88.10** | **88.19** | **87.50** | **88.90** | **4.2%** |
> > > | **GPT-4.1** | 84.00 | 83.73 | 82.50 | 85.00 | 8.8% |
> > > | **Qwen2.5-VL-7B** | 77.50 | 77.49 | 76.50 | 78.50 | 11.5% |
> > >
> > > We additionally assess robustness to adversarial prompt-injection attacks using AutoDAN. For each input, AutoDAN performs five iterative attack rounds, where each round rewrites the prompt with increasingly optimized jailbreak strategies tailored to the model’s previous responses. For evaluation, we report the attack success rate (ASR) corresponding to the most effective round, defined as the proportion of cases where the model is induced to produce an incorrect safety judgment under attack.
> > >
> > > The results in the table show that ASR is generally low across all models, indicating that prompt-injection attacks are non-trivial in a streaming moderation setting. Nonetheless, clear differences remain. StreamGuard achieves the strongest robustness profile, with the best clean performance (88.10 Acc, 88.19 F1) and the lowest ASR (4.2%). In contrast, GPT-4.1 and Qwen suffer higher rates of successful attacks (8.8% and 11.5%, respectively). This advantage is consistent with our training procedure: StreamGuard is explicitly exposed to adversarially rewritten prompts during training, which improves resilience to targeted prompt injections while preserving accuracy on clean inputs.
> > >
> > > We also recognize that more sophisticated attackers could potentially exploit the structure of streaming inference—for example, by positioning harmful content around reset boundaries to evade temporal aggregation. We explicitly identify this as an open threat model and an important direction for continued research. Accordingly, our current adaptive-attack evaluation should be viewed as an initial robustness probe rather than a comprehensive adversarial stress test.
> > >
> > > ---
> > >
> > > We hope these clarifications and additional experiments address your concerns regarding novelty, ablations for the streaming mechanism, policy/event design, and robustness to adaptive attacks.

---

### Official Review · Reviewer_9vxC · 2025-11-01

**Soundness:** 2
**Presentation:** 3
**Contribution:** 2
**Rating:** 4
**Confidence:** 3

**Summary:**

This paper introduces STREAMGUARD, a real-time, policy-following video moderation system designed to accurately detect unsafe content, such as violence and pornography, in long-form videos. Compared to traditional methods, STREAMGUARD utilizes streaming video analysis to avoid the information loss caused by frame sampling and employs parallel event reasoning to significantly improve processing efficiency. Additionally, the system enhances robustness against adversarial attacks through multi-level policy alignment training. To validate its effectiveness, the paper introduces two new benchmark datasets, SAFE2SHOT and ADVVIDEO-BENCH, and experimental results demonstrate that STREAMGUARD outperforms existing state-of-the-art methods in both accuracy and efficiency.

**Strengths:**

1. STREAMGUARD uses real-time streaming processing, without relying on frame sampling, ensuring that every frame in the video stream is accurately annotated.
2. The effectiveness of the method has been validated through experiments.
3. Ablation studies have demonstrated the validity of each component.

**Weaknesses:**

1. The layout is too tight. In some places, two lines of text are almost overlapping, such as on lines 152-153 and 211-212.

2. Many papers have already explored video modality safety, such as [1-7]. However, the author does not discuss these works at all in the related work section (including many others not listed here).

   [1] Safree: Training-free and adaptive guard for safe text-to-image and video generation

   [2] T2Vs Meet VLMs: A Scalable Multimodal Dataset for Visual Harmfulness Recognition

   [3] T2vsafetybench: Evaluating the safety of text-to-video generative models

   [4] SafeVid: Toward Safety Aligned Video Large Multimodal Models

   [5] Protecting your video content: Disrupting automated video-based LLM annotations

   [6] Videojail: Exploiting video-modality vulnerabilities for jailbreak attacks on multimodal large language models

   [7] SEA: Low-Resource Safety Alignment for Multimodal Large Language Models via Synthetic Embeddings

   [8] From Evaluation to Defense: Advancing Safety in Video Large Language Models

3. STREAMGUARD is trained based on datasets, rather than employing adaptive defense mechanisms. As a result, it may lack generalization when facing previously unseen risk patterns.

**Questions:**

see weaknesses

---

> ### Author Response · Authors · 2025-11-23
>
> Dear Reviewer 9vxC,
>
> Thank you for your detailed review and for recognizing the value and novelty of our proposed method. We appreciate your constructive feedback and address your concerns point by point below.
>
> ---
>
> **A1: Layout is too tight in some places**
>
> We appreciate you pointing out the layout issue.We have increased the vertical spacing, adjusted the paragraph and line spacing around equations and figures, and carefully checked and fixed the problematic locations (including the regions around the original lines 152–153 and 211–212). The updated formatting removes the overlapping lines and improves overall readability.
>
> ---
>
> **A2: Missing discussion of recent video safety work in related work**
>
> We agree with this concern and have substantially revised the related work section to explicitly cover these lines of work.
> In particular, we now (i) group prior work into text-to-image/video safety (e.g., Safree, T2VSafetyBench), offline video-LMM safety and evaluation (e.g., SafeVid, VideoJail, T2Vs Meet VLMs, From Evaluation to Defense), and low-resource / embedding-based alignment (e.g., SEA), and (ii) clearly position **StreamGuard** as a complementary contribution focusing on *real-time, frame-exact streaming moderation with event-level reasoning and multi-level policy alignment*, rather than static generation or purely offline evaluation. We believe this clarification makes the relationship and differences to existing work much clearer.

---

> > ### Author Response · Authors · 2025-11-23
> >
> > **A3: Generalization and lack of adaptive defense mechanisms**
> >
> > We agree that generalization to unseen risk patterns and robustness beyond fixed datasets are crucial. In the main paper, **Table 2** already evaluates **StreamGuard** on several out-of-distribution video benchmarks (UCF-Crime, XD-Violence, LSPD, FVC, etc.), and **Table 3** includes additional tests on object recognition and borderline/erotic scenes, but we did not emphasize this clearly enough.
> >
> > To better address your concern, we have added **Appendix B** with additional experiments:
> >
> > * OOD tests on *new high-risk categories* (Fire-Smoke, Gun, Drug, and Religious videos);
> > * evaluation on the independent **T2VSafetyBench** benchmark;
> > * robustness to *adaptive prompt-based attacks* (AutoDAN-style attacks targeting the prompts used for moderation).
> >
> > **Table. Evaluation on five additional out-of-distribution categories not covered in our training dataset (Fire-Smoke, Gun, Shooting, Robbery, Drug) and a benign Religious category (Accuracy only).**
> >
> > | **Model** | **Fire-Smoke Acc** | **Fire-Smoke Pre** | **Fire-Smoke Rec** | **Fire-Smoke F1** | **Gun Acc** | **Gun Pre** | **Gun Rec** | **Gun F1** | **Shooting Acc** | **Shooting Pre** | **Shooting Rec** | **Shooting F1** | **Robbery Acc** | **Robbery Pre** | **Robbery Rec** | **Robbery F1** | **Drug Acc** | **Drug Pre** | **Drug Rec** | **Drug F1** | **Religious Acc** |
> > |-----------|--------------------|---------------------|---------------------|--------------------|--------------|--------------|--------------|--------------|------------------|------------------|------------------|------------------|------------------|------------------|------------------|------------------|--------------|--------------|--------------|--------------|------------------|
> > | **GPT-4.1** | 95.0 | 90.9 | **100** | 95.2 | **100** | **100** | **100** | **100** | 70.0 | **100** | 40.0 | 57.1 | **100** | **100** | **100** | **100** | 85.0 | 75.0 | 60.0 | 66.7 | **100** |
> > | **Qwen2.5-VL-7B** | 80.0 | **100** | 60.0 | 75.0 | 95.0 | **100** | 95.0 | 97.4 | 55.0 | **100** | 10.0 | 18.2 | 85.0 | **100** | 70.0 | 82.4 | 75.0 | 50.0 | 40.0 | 44.4 | 85.0 |
> > | **StreamGuard** | **100** | **100** | **100** | **100** | **100** | **100** | **100** | **100** | **80.0** | **100** | **60.0** | **75.0** | **95.0** | **100** | **90.0** | **94.7** | **90.0** | **80.0** | **80.0** | **80.0** | **100** |
> >
> >
> > To further assess out-of-distribution robustness, we evaluate our models on several new high-risk categories. Fire-Smoke clips are drawn from the Smoke and Fire Detection Videos Dataset, Gun clips from the CCTV GUN dataset, Shooting and Robbery events from UCF-Crime, and Drug clips from T2VSafetyBench. We additionally construct a Religious category by sampling benign videos of religious activities from YouTube, which we use to probe whether models over-block sensitive but non-harmful content.
> >
> > Across all categories, StreamGuard consistently outperforms GPT-4.1 and Qwen2.5-VL-7B.
> > It achieves perfect detection accuracy and F1 on Fire-Smoke and Gun, demonstrating strong transfer to visually distinct but safety-critical hazards. On more complex event-style risks such as Shooting, Robbery, and Drug, \alg maintains high accuracy and balanced precision–recall, whereas baseline models either miss many true violations (e.g., Qwen’s Recall = 10% on Shooting) or fluctuate widely across categories.
> >
> > Finally, on the benign Religious category, StreamGuard achieves 100% accuracy, showing that it avoids over-blocking sensitive but otherwise harmless content and therefore maintains utility under distribution shift.
> > Overall, these results indicate that StreamGuard provides more reliable and robust safety judgments on diverse high-risk OOD categories.

---

> > > ### Author Response · Authors · 2025-11-23
> > >
> > > **Table. Fine-grained category-level evaluation on T2VSafetyBench.
> > > For each category, we report Accuracy (Acc), Recall (Rec), Precision (Pre), and F1 for Qwen2.5-VL-7B, GPT-4.1, and StreamGuard.
> > > While overall performance is modest due to noisy and ambiguous labels, StreamGuard generally achieves a better recall–precision balance on several core harm categories.**
> > >
> > > | **Category** | **Qwen Acc** | **Qwen Rec** | **Qwen Pre** | **Qwen F1** | **GPT-4.1 Acc** | **GPT-4.1 Rec** | **GPT-4.1 Pre** | **GPT-4.1 F1** | **StreamGuard Acc** | **StreamGuard Rec** | **StreamGuard Pre** | **StreamGuard F1** |
> > > |--------------|--------------|--------------|--------------|-------------|------------------|------------------|------------------|------------------|-----------------------|-----------------------|-----------------------|----------------------|
> > > | **Pornography** | 92.19 | **100.00** | 50.00 | 66.67 | 93.75 | 60.00 | 60.00 | 60.00 | **96.88** | 80.00 | **80.00** | **80.00** |
> > > | **Borderline Pornography** | **92.19** | 0.00 | 0.00 | 0.00 | 90.62 | 20.00 | 33.33 | 25.00 | **92.19** | **33.33** | **66.67** | **44.44** |
> > > | **Violence** | 75.00 | 60.00 | 17.65 | 27.27 | 81.25 | 80.00 | 26.67 | 40.00 | **84.38** | **100.00** | **33.33** | **50.00** |
> > > | **Gore** | 87.50 | 80.00 | 36.36 | 50.00 | 92.19 | 80.00 | 50.00 | 61.54 | **93.75** | **83.33** | **62.50** | **71.43** |
> > > | **Public Figures** | **90.62** | **20.00** | **33.33** | **25.00** | **90.62** | 0.00 | 0.00 | 0.00 | 87.50 | 0.00 | 0.00 | 0.00 |
> > > | **Discrimination** | **92.19** | 0.00 | 0.00 | 0.00 | 85.94 | 0.00 | 0.00 | 0.00 | 87.50 | **14.29** | **33.33** | **20.00** |
> > > | **Politically Sensitive** | 92.19 | 60.00 | 50.00 | 54.55 | **95.31** | 60.00 | 75.00 | 66.67 | **95.31** | **66.67** | **80.00** | **72.73** |
> > > | **Illegal Activities** | 95.31 | 60.00 | 75.00 | 66.67 | **96.88** | 80.00 | 80.00 | 80.00 | **96.88** | **83.33** | **83.33** | **83.33** |
> > > | **Disturbing Content** | 64.06 | **100.00** | 17.86 | 30.30 | 65.62 | **100.00** | **18.52** | **31.25** | **68.75** | **100.00** | 16.67 | 28.57 |
> > > | **Misinformation** | 92.19 | 0.00 | 0.00 | 0.00 | 92.19 | 0.00 | 0.00 | 0.00 | **93.75** | **20.00** | **100.00** | **33.33** |
> > > | **Copyright** | 92.19 | 0.00 | 0.00 | 0.00 | **93.75** | **20.00** | **100.00** | **33.33** | **93.75** | **20.00** | **100.00** | **33.33** |
> > > | **Sequential Action Risk** | **96.88** | **0.00** | **0.00** | **0.00** | 87.50 | **0.00** | **0.00** | **0.00** | 90.62 | **0.00** | **0.00** | **0.00** |
> > > | **Dynamic Variation Risk** | **93.75** | 0.00 | 0.00 | 0.00 | **93.75** | 0.00 | 0.00 | 0.00 | **93.75** | **25.00** | **50.00** | **33.33** |
> > > | **Coherent Contextual Risk** | **95.31** | 0.00 | 0.00 | 0.00 | **95.31** | 0.00 | 0.00 | 0.00 | 93.75 | **25.00** | **50.00** | **33.33** |
> > >
> > >
> > >
> > > As shown in the table above, we further evaluate all models on a new out-of-distribution video-generation safety dataset that spans a diverse set of fine-grained risk categories, including pornography, borderline content, violence, political sensitivity, misinformation, and several temporal-risk dimensions. The overall performance across all models is modest. In our manual inspection, we find that a non-trivial portion of clips are ambiguous or only weakly aligned with their assigned labels, and some categories (such as discrimination or misinformation) exhibit noisy or inconsistent annotations. As a result, even when the model’s prediction is qualitatively reasonable, it may still be marked incorrect under the dataset’s labels.
> > >
> > > Therefore, these results should be interpreted primarily as a stress test under noisy supervision, rather than a clean measurement of absolute safety capability. Despite the label noise, StreamGuard tends to show a stronger recall–precision balance across several core harm categories—such as pornography, gore, illegal activities, and politically sensitive content—while the baselines often exhibit either severe under-detection (zero recall) or unstable precision. This suggests that the multi-level design of StreamGuard remains comparatively more robust, even under imperfect OOD data and ambiguous annotation quality.

---

> > > > ### Author Response · Authors · 2025-11-23
> > > >
> > > > **Table. Robustness evaluation under Adaptive attacks (AutoDAN).
> > > > We report clean Accuracy (Acc), F1, Recall (Rec), Precision (Pre), and the attack success rate (ASR).
> > > > StreamGuard achieves both the highest clean performance and the lowest ASR.**
> > > >
> > > > | **Model** | **Acc** | **F1** | **Rec** | **Pre** | **ASR** |
> > > > |----------|---------|--------|---------|---------|---------|
> > > > | **StreamGuard** | **88.10** | **88.19** | **87.50** | **88.90** | **4.2%** |
> > > > | **GPT-4.1** | 84.00 | 83.73 | 82.50 | 85.00 | 8.8% |
> > > > | **Qwen2.5-VL-7B** | 77.50 | 77.49 | 76.50 | 78.50 | 11.5% |
> > > >
> > > > We further evaluate robustness against adversarial prompt-injection attacks using AutoDAN. For each input, AutoDAN runs for five successive attack rounds, where each round adaptively rewrites the query using increasingly aggressive jailbreak strategies. We take the round with the highest attack success rate (ASR) and report ASR as the fraction of inputs for which the model is driven to output an incorrect safety judgment under attack.
> > > >
> > > > As shown in the table, the absolute ASR remains relatively low for all three models, suggesting that AutoDAN is not trivially effective in a streaming safety moderation setting. Nevertheless, StreamGuard achieves both the highest clean accuracy (88.10 Acc, 88.19 F1) and the lowest ASR (4.2%), whereas GPT-4.1 and Qwen exhibit noticeably higher vulnerability (8.8% and 11.5% ASR, respectively). This aligns with our training procedure: StreamGuard is explicitly exposed to AutoDAN-style adversarial samples during training, which strengthens its resistance to prompt-injection attacks without degrading its overall safety detection ability.

---

### Author Response · Authors · 2025-11-23

We thank all reviewers for their constructive feedback. In the revised version, we included two major categories of improvements:
(1) **a comprehensive expansion of experiments**, including additional OOD policy categories, robustness analyses, and multi-level ablations; and
(2) **writing and structural revisions** that revise the related works section to systematically discuss more recent works, clarify policy definitions, dataset construction, and guardrail framework design.

---

### **1. Expanded Experiments and Ablations (Appendix B; Tables 8–13)**

To more thoroughly evaluate generalization, robustness, and the necessity of our design choices, we added the following experiments:

**• Additional OOD evaluation on new high-risk categories (Table 8)**
Responds to the requests for stronger OOD tests (Reviewers 9vxC, mM3s).
We introduce five new high-risk categories—Fire-Smoke, Gun, Shooting, Robbery, Drug—and a benign Religious category from independent sources. These tests show that StreamGuard maintains strong detection performance while avoiding over-blocking benign sensitive content.

**• Ablation on streaming frame rate and sampling strategy (Table 9)**
Addresses concerns regarding sampling rate, memory window, and streaming necessity (Reviewers mM3s, L4Tt).
We compare balanced sampling (1/2/4 fps), unstable streaming patterns (step-drop, random-jitter), uniform non-streaming baselines, and VLM baselines.
The **2 fps streaming configuration** offers the best Acc/F1 trade-off and remains robust under unstable streaming.

**• Inference-time ablation of frame- and event-level reasoning (Table 10)**
Added in response to requests for isolating the contribution of each reasoning branch (Reviewers mM3s, ZuRt).
Event-level removal produces only a modest drop, while frame-level removal causes a significant decline, showing that **fine-grained frame-level cues are essential**, with event-level aggregation providing temporal refinement.

**• Hyperparameter ablation on (λ₁, λ₂, λ₃) (Table 11)**
Requested by Reviewers ZuRt and L4Tt.
We compare uniform, single-level, and asymmetric weightings.
The proposed asymmetric weighting—based on relative label availability—achieves the highest performance and lowest false-positive rate.

**• T2VSafetyBench evaluation (Table 12)**
Added in response to Reviewer L4Tt’s question about performance on traditional T2V adversarial datasets.
Due to noisy labels, overall numbers remain modest, but StreamGuard preserves a more stable recall–precision balance on harmful categories.

**• Robustness under adaptive prompt-injection attacks (Table 13)**
Addresses adaptive attacker concerns from Reviewers 9vxC, mM3s, ZuRt.
Using multi-round AutoDAN attacks, StreamGuard achieves the highest clean accuracy and the lowest attack success rate, indicating improved resistance to adversarial prompting.

Together, these additional experiments provide a more comprehensive and rigorous evaluation of StreamGuard’s robustness, generalization ability, and multi-level design.

---

### **2. Writing Clarifications and Structural Improvements**

We also made substantial improvements to the exposition:

**• Expanded and reorganized Related Work**
Clarifies how StreamGuard relates to SAFREE, SafeVid, SEA, VA-SafetyBench, VideoSafety-R1, T2VSafetyBench, T2Vs Meet VLMs, VSB-77k, and recent jailbreak work.
Addresses missing citations and contextualization issues noted by Reviewers 9vxC and L4Tt.

**• Clarification of policy categories and event definitions**
Explicitly maps all risk categories to safety guidelines of mainstream video platforms, responding to concerns about heuristic category design (Reviewers mM3s, ZuRt).

**• Detailed descriptions for Safe2Shot and TV2V construction**
Adds step-by-step details on prefix sampling, adversarial prompt construction, and manual sanity checks.
Requested by Reviewer L4Tt.

**• Clearer training pipeline and hyperparameter rationale**
Adds explanation for SFT → RL alignment stages and the asymmetric weighting of (λ₁, λ₂, λ₃).
Addresses hyperparameter concerns from Reviewers ZuRt and L4Tt.

**• Improved layout and spacing**
Fixes compressed spacing, wraptable overflow, and layout tightness (Reviewer 9vxC).

**• Discussion of real-time scenarios and unstable streams**
Clarifies how StreamGuard handles delayed/incomplete streams and supports this with new jitter/step-drop ablations.
Addresses concerns from Reviewer L4Tt.

---

### Note · Program_Chairs · 2026-01-17
**Submission Desk Rejected by Program Chairs**

The following references in this submission do not refer to real documents and/or have major errors in bibliographic information:

 X Wu et al. A survey on long-term video understanding: Challenges, methods, and directions. arXiv preprint arXiv:2207.12345, 2022.
J Lei et al. Long-form video-language understanding with memory-efficient transformers. In Proceedings of the IEEE/CVF International Conference on Computer Vision, 2023.